# Optimal control of multiple myeloma assuming drug resistance and off-target effects

James G. Lefevre [ID][1,2][*], Brodie A. J. Lawson [ID][2,3], Pamela M. Burrage [ID][2,3], Diane M. Donovan[1,2], Kevin Burrage[2,3,4]

**1** School of Mathematics and Physics, The University of Queensland, Brisbane, Queensland, Australia, **2** ARC Centre of Excellence, Plant Success in Nature and Agriculture, St. Lucia, Queensland, Australia, **3** School of Mathematical Sciences, Queensland University of Technology, Brisbane, Queensland, Australia, **4** Department of Computer Science, University of Oxford, Oxford, United Kingdom

* j.lefevre@uq.edu.au

**Data availability statement:** There are no primary data in the paper; all code is available on a GitHub repository at https://github.com/jameslefevre/dara-optimal-control/tree/main.

## Abstract

Multiple myeloma (MM) is a plasma cell cancer that occurs in the bone marrow. A leading treatment for MM is the monoclonal antibody Daratumumab, targeting the CD38 receptor, which is highly overexpressed in myeloma cells. In this work we model drug resistance via loss of CD38 expression, which is a proposed mechanism of resistance to Daratumumab treatment. We develop an ODE model that includes drug resistance via two mechanisms: a direct effect in which CD38 expression is lost without cell death in response to Daratumumab, and an indirect effect in which CD38 expression switches on and off in the cancer cells; myeloma cells that do not express CD38 have lower fitness but are shielded from the drug action. The model also incorporates competition with healthy cells, death of healthy cells due to off-target drug effects, and a Michaelis-Menten type immune response. Using optimal control theory, we study the effect of the drug resistance mechanisms and the off-target drug effect on the optimal treatment regime. We identify a general increase in the duration and costs of optimal treatment, as a result of these added mechanisms. Several distinct optimal treatment regimes are identified within the parameter space.

## Author summary

In this work we investigate a model of multiple myeloma, a cancer of the bone marrow, and its treatment with the drug Daratumumab. The model incorporates proposed mechanisms by which the cancer resists Daratumumab by reduced expression of the receptor CD38, which is the drug target and normally abundant in the cancer cells. The model includes an off-target effect, meaning that the drug treatment destroys some healthy cells alongside the targeted cancer cells. Both mechanisms can reasonably be expected

**Funding:** This work was supported in part by Australian Research Council Centre of Excellence for Plant Success in Nature and Agriculture (CE200100015). DMD and KB are CIs in this centre, which funds the positions of JGL and BAJL. The funders had no role in study design, data collection and analysis, decision to publish, or preparation of the manuscript.

**Competing interests:** The authors have declared that no competing interests exist.

to reduce the efficacy of the drug. We investigate the model using optimal control methods, which are used to find the drug dose over time which best balances the financial and health costs of treatment against cancer persistence, according to a specified cost function. We show that this drug resistance and off-target effect prolongs the optimal treatment and increases the burden of both the disease and drug. We analyse the distinct effects of the controlling parameters on each of these costs factors as well as the time course, and identify conditions under which extended treatment is required, with either intermittent treatment or a steady reduced dose. Extended treatment may be indefinite or for a fixed period.

## Introduction

Myeloma is a plasma cell cancer that occurs in the bone marrow. Myeloma cells typically form masses of cancerous tissue, and the disease is known as multiple myeloma when more than one mass is present. Myeloma can crowd out healthy marrow tissue, leading to a range of potential deficiencies, and invade and weaken bone. It may also cause damage via production of abnormal antibodies. The burden of morbidity and mortality is substantial, with an estimated 187,952 cases and 121,388 deaths globally in 2022, ranking 21st and 17th respectively amongst cancer sites [1]. A number of treatment options are available, although a complete cure has proved elusive [2,3]. This often results in multiple lines of therapy being used in succession, leading to high costs. A 2022 United States study considered patients with refractory multiple myeloma after progressing through at least 4 lines of therapy, and found average total treatment costs of US$670,561 over an average 21.9 month follow-up period [4].

In general, myeloma cells are marked by very high CD38 expression, motivating the use of the monoclonal antibody Daratumumab (Dara), which effectively targets myeloma via several mechanisms [5]. However, CD38 is also expressed in a wide range of cell types, resulting in important and complex off-target effects [6]. Dara is a leading treatment for multiple myeloma, commonly sold under the brand name Darzalez. We note that several other drugs have been developed to treat myeloma, such as Elotuzumab [7,8] and Lenalidomide [9], which can be used together in combination with the adjunct drug Dexamethasone as a combination treatment for refractory disease [10]. More recently combination therapies using Dara have also been developed [11]. However, in this work we consider treatment using Dara only. Dara monotherapy has primarily been used for heavily pretreated relapsed or refractory disease [5], although it has shown efficacy in preventing the precursor condition, smouldering multiple myeloma, from progressing [12].

Myeloma develops resistance to Dara. The dynamics are not fully understood, although various mechanisms have been proposed and combination therapies and recurrent treatment have had clinical success [13]. In this work we focus on one known resistance mechanism, in which myeloma cells resist Dara via loss of CD38 expression. This may occur via Darwinian selection, where a differential death rate under Dara treatment leads to a relative increase in myeloma cells with low CD38 expression. There may also be a direct loss of expression (without cell death) in response to drug exposure, as has been shown to occur in red blood cells [14]. This loss of expression is assumed to persist in daughter cells, as in Lamarckian evolution. Importantly, loss of CD38 expression does not lead to the final failure of Dara, and the expression can recover.

Using optimal control theory, we investigate how this drug resistance affects treatment protocols for Dara that balance the cost of treatment with the burden of disease. The cost of

treatment includes health impacts as well as the financial cost of the drug and its administration. We also include an off-target effect in our model, in which the drug causes mortality in the surrounding population of healthy cells. We find in general that these effects support a more prolonged treatment regime and increase overall costs. We further investigate the connection between the various dynamics of the model and the total cost and duration of treatment. Notably, we find that with a linear cost function, the optimal drug dosage over time can have distinct functional forms depending on parameter values. An initial period of maximal dosage may be followed by lower level treatment at constant or decreasing dose, possibly after a pause. In certain cases where a more prolonged or indefinite treatment is required, we find that a regular intermittent treatment regime is optimal. This may help to inform maintenance Dara treatment, which has shown success preventing relapse when used as a reduced frequency monotherapy following the primary combination therapy [15].

## Dynamical systems and optimal control theory

Biological applications of dynamical systems models were pioneered with the logistic model of Verhulst [16,17], representing exponential population increase constrained by a maximum carrying capacity. The famous Lotka–Volterra predator-prey model was first used to study interacting chemical species [18], then later applied to an ecological model, showing that the interactions between a prey species and a predator species could produce a continuing oscillation of populations over time [19].

Cancer biology typically involves complex interactions of cancer cells with their microenvironment and with a range of immune and other cell types. A range of dynamical systems models have been developed to help understand this clinically critical biology [20]. State variables represent populations of cells and other relevant species. A number of papers (e.g. [21, 22]) have modelled cancer-immune interactions through a predator-prey framing, with cancer cells as the prey and cytotoxic T-cells, a type of white blood cell which destroy diseased cells, acting as the predator. Modelling of this interaction is of particular interest due to the introduction of CAR T-cell therapy [23], which relies on modified T-cells with an increased capacity to target cancer. A limitation of the predator-prey analogy is that consumption of prey strengthens the predator, whereas in cancer the first-order effect of immune cell "predation" weakens the immune cell population. However, positive feedback may be produced via various second order effects. The appropriate approaches for modelling these complex interactions is the subject of active research [24,25].

A *control* is an exogenous variable $u(t)$ incorporated into a dynamical system:

$$\frac{d\mathbf{x}}{dt} = \mathbf{f}(\mathbf{x}, u).$$

$$(1)$$

An *optimal control* minimises some cost function defined on $u$ and the state variables $\mathbf{x}(t)$ over a specified time window $[t_0, t_f]$, for a given initial state:

$$\mathbf{x}(t_0) = \mathbf{x_0}.$$

$$(2)$$

Since the control may vary freely over the time window, determining the optimal control is, in general, challenging. Specialised numerical methods are required, with mathematical and numerical constraints that restrict the form of the cost function. Our approach is based on Pontryagin's maximum principle [26]. We note that this approach does not guarantee a

global optimum. The cost function must have the form

$$J = \underbrace{\phi(\mathbf{x}(t_f))}_{\text{end state cost}} + \int_{t_0}^{t_f} \underbrace{\mathcal{L}(t, \mathbf{x}(t), u(t))}_{\text{ongoing cost}} \, dt. \tag{3}$$

If $\mathcal{L}$ and $\mathbf{f}$ both have a linear dependency on $u$, then bounds must be imposed on $u$ in order to give a well-defined solution. The optimal control often takes the form of a step function. If $\mathcal{L}$ is a convex function of $u$ the problem is generally more tractable, giving a smoothly varying optimal control without the need to impose bounds. We will consider cost functions of both types (see Eqs (7) and (8)).

Optimal control theory has been applied to clinical models to find theoretically optimal treatment regimes, with controls representing drug dose levels over time and cost functions designed to balance the monetary and health cost of treatment against the burden of disease. Important recent work includes applications to cancer immunotherapy, including generalised Lotka-Volterra predator-prey models [27] and models of combination therapy [28].

In this work we take a different approach, incorporating a very simplified immune response and placing focus instead on the drug resistance mechanism and off-target effects discussed above. See [29] for a detailed mathematical treatment of immune dynamics in multiple myeloma.

## Model of myeloma and Daratumumab

Crowell et al. [30] proposed a dynamical system model of acute myeloid leukaemia incorporating a competition between healthy and cancerous cells for space in the marrow, with proliferation of both populations restricted as the total cell population approaches the carrying capacity. The model features the migration of healthy cells into the compartment from a separate stem cell compartment, and migration of both healthy and cancerous cells into the blood system.

Sharp et al. [31] applied an optimal control methodology to a modified version of the Crowell model, with the addition of an immune response to cancer. The immune response was represented using a Michaelis-Menten term, which models a bounded immune capacity that initially scales with the cancer level but has a maximum capacity to remove cancer cells; this has the effect, for appropriate parameter choices, of allowing stable steady states with and without cancer present. This modification allows finite term treatment to result in a permanent control of the cancer; the authors found that this property was required for convergence of the optimal control algorithm. These works provide a model that supports the expected dynamics of cancer and cancer treatment, as well as a proven methodology for obtaining controls using both linear and quadratic cost functions.

We develop a dynamical systems model of myeloma that adapts the core features of the Sharp model [31]. Our model also contains healthy and cancerous populations of marrow cells that compete for space, while the downstream blood cell populations are omitted, as they do not affect the marrow cells or the cost function. The upstream stem cell population is modelled implicitly as an influx of healthy cells into the marrow (rate $\beta_A$), supporting the healthy marrow population $A$. There is an implicit assumption that the stem cell population is at steady state; this causes only a transitory divergence from the Sharp model, and has no impact on the optimal control results as they assume steady state at $t = 0$.

The drug resistance mechanism requires that cancer cell CD38 expression can vary, and the Darwinian selection mechanism requires natural variation in the population. We do not know the specifics of this distribution, so we assumed a simplified binary model. We replace

the cancer population with CD38+ and CD38- cancer cell subpopulations $P$ and $N$, of which only $P$ is susceptible to the drug. These represent alternate states of a single population, so it is assumed that cells move between $P$ and $N$ at rates $\delta_P$ and $\delta_N$. We will refer to this mechanism as expression switching. Given the general overexpression of CD38 in myeloma, we assume that $\delta_N = 10\delta_P$ (Sect B in S2 Text suggests limited sensitivity to the exact ratio), and that fitness is significantly lower in $N$. It has been shown that CD38 is a growth and survival factor in a lung cancer model system [5], and so it is reasonable to assume that $N$ has both lower proliferation and higher death rate than $P$. We also allow for both direct drug-induced loss of CD38 expression ($\delta_{Pu}$), and an off-target effect modelled by drug-induced death of healthy cells ($\mu_{Au}$). Note that off-target drug effects can be modelled implicitly in the cost function, but this explicit approach accounts for an interaction with population dynamics; in particular, a depleted healthy population may reduce space competition and allow the cancer population to recover faster.

The complete model (Fig 1) is as follows, where the three state variables $\mathbf{x} = (A, P, N)$ are each expressed as a proportion of the marrow carrying capacity:

$$\frac{dA}{dt} = \beta_A + \rho_A A(1 - A - N - P) - \mu_A A - \mu_{Au} u A \tag{4}$$

$$\frac{dP}{dt} = \rho_P P(1 - A - N - P)$$

$$-\delta_P P + \delta_N N - \delta_{Pu} u P - \mu_P P - \mu_{Pu} u P - \frac{\alpha P}{\gamma + P + N} \tag{5}$$

$$\frac{dN}{dt} = \rho_N N(1 - A - N - P) + \delta_P P - \delta_N N + \delta_{Pu} u P - \mu_N N - \frac{\alpha N}{\gamma + P + N} \tag{6}$$

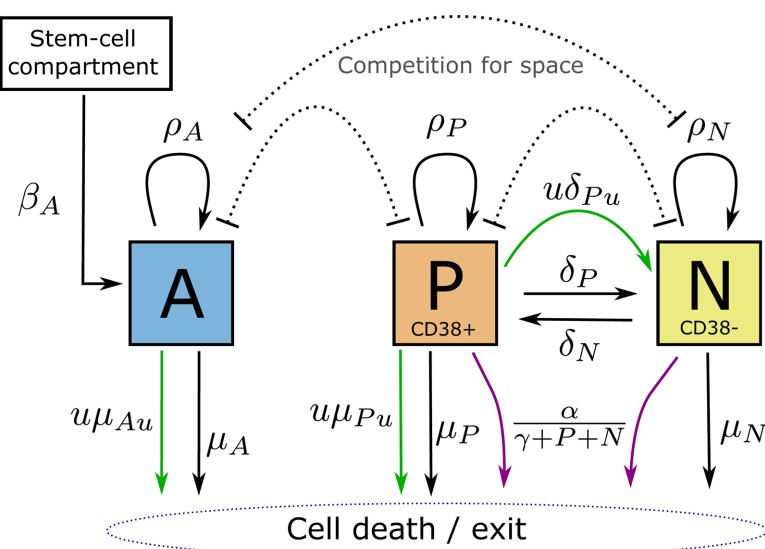

**Fig 1. Model of multiple myeloma treatment with Dara (control, $u$), including an immune response (purple arrows) and drug resistance mechanisms via loss of CD38 expression.** The cell populations are healthy marrow cells ($A$), CD38+ cancer ($P$) and CD38- cancer ($N$), and all arrows represent movement of cells between populations or states. Three drug actions are included (green arrows): cell mortality and loss of CD38 expression in the CD38+ cancer cells, and off-target cell mortality in healthy cells within the compartment.

Here the control $u \geq 0$ represents the dosage rate of Dara. The state variables must also be non-negative to be physically realisable. A description of the model parameters, and the default values used, are listed in Table 1. In the column on the right-hand side we list a second set of parameter values which, together with the condition $N(0)$, define the *Null-N* model. This simplification reproduces the core features of the Sharp model and is used as a negative control.

The non-zero parameters of the Null-N model were adapted from the Sharp model (Table 1 of [31]), which were selected to produce balanced dynamics supporting both healthy and cancerous states and the capacity for effective drug control. Our parameters $\mu_A$, $\rho_P$ and $\mu_P$ correspond to $\delta_A$, $\rho_L$ and $\delta_L$ respectively in the Sharp model, $\beta_A = \delta_S(1 - \delta_S/\rho_S)$ is the exit rate from the $S$ compartment at steady state, and the remaining parameters retain the same name. These values are also used in the full model, except for a small adjustment to $\rho_P$ and $\mu_P$ to balance the introduction of $N$ with substantially lower proliferation and higher mortality. The additional parameters in the full model were described previously, other than the Dara related parameters which we consider now. The effect of a unit of control on the mortality rate of CD38+ cancer cells, $\mu_{Pu}$, is fixed at one; this defines the scale for the control $u$. Since CD38 is typically highly overexpressed in myeloma, it can be assumed that $\mu_{Au}$ is substantially lower than $\mu_{Pu} = 1$; we use 0.1 by default, although higher values are also considered. We also choose a conservative initial value of $\delta_{Pu} = 0.2$, implying the direct loss of expression from Dara is a smaller effect than mortality, but with higher values considered. Note that in our model, as in the Sharp and Crowell models, the unit of time is abstract and parameters are not calibrated to real data.

## Cost functions

An optimal control can be calculated only with reference to a cost function. This function encodes the health cost of cancer presence, as well as the cost of the drug dose over time; the latter comprises both its direct financial cost and health effects due to its side effects. The most appropriate mapping between these factors and cost is not obvious.

In the absence of specific information, a simple linear combination of the cost factors is the most natural choice for the cost function. Quadratic functions are also widely used, with the justification that super-linear costs are evident in many contexts. This may also be due

**Table 1. Model parameters with descriptions and default values. The right-hand side column gives the parameter values for a simplified model (Null-N).**

| Description | Parameter | Value | Null-N |
|---|---|---|---|
| Influx of healthy cells | $\beta_A$ | 0.1008 | 0.1008 |
| Proliferation rate of $A$ (healthy cells) | $\rho_A$ | 0.43 | 0.43 |
| Rate of death or other exit for $A$ | $\mu_A$ | 0.44 | 0.44 |
| Off-target mortality effect on $A$ per unit of Dara (control) | $\mu_{Au}$ | 0.1 | 0 |
| Proliferation rate of $P$ (CD38+ myeloma) | $\rho_P$ | 0.28 | 0.27 |
| Rate of death or other exit for $P$ | $\mu_P$ | 0.048 | 0.05 |
| Additional death rate of $P$ per unit of Dara (control) | $\mu_{Pu}$ | 1 | 1 |
| Proliferation rate of $N$ (CD38- myeloma) | $\rho_N$ | 0.15 | 0 |
| Rate of death or other exit for $N$ | $\mu_N$ | 0.06 | 0 |
| Rate of loss in CD38 expression from $P$ (to $N$) | $\delta_P$ | 0.003 | 0 |
| Rate of gain in CD38 expression from $N$ (to $P$) | $\delta_N$ | 0.03 | 0 |
| Increased loss in CD38 expression in $P$ per unit of Dara | $\delta_{Pu}$ | 0.2 | 0 |
| Immune control rate | $\alpha$ | 0.015 | 0.015 |
| Immune control half saturation | $\gamma$ | 0.1 | 0.1 |

in large part to their superior mathematical and numerical tractability compared with linear functions [32]. We consider two optimal control cost functions, linear and quadratic in form (7, 8), to provide insight into the influence of the cost assumptions and the robustness of any conclusions. In each case the cost function takes the form of (3) with $\phi(\mathbf{x}(t_f)) = 0$. The health cost due to cancer is assumed to depend on the total cancer population, $P + N$.

$$\text{Linear cost function:} \qquad \mathcal{L} = u + P + N, \text{ where } 0 \le u \le 1. \qquad (7)$$

$$\text{Quadratic cost function:} \qquad \mathcal{L} = u^2 + (P + N)^2. \qquad (8)$$

Both cost functions consist of a simple unweighted sum of control and cancer cost terms. It is reasonable to ask whether different weightings affect the form of the optimal control (see [33], for example), and in Sect C in S2 Text we perform a sensitivity analysis in which we consider altered weights. Thus our sensitivity analysis also considers a general super-linear cost function containing both linear and quadratic terms, to investigate whether results derived from this 'mixed' cost function are broadly intermediate between those generated using purely linear or purely quadratic cost functions.

The cost function cannot take into account the system state outside the selected time window (for example, cancer recurrence). The function $\phi(\mathbf{x}(t_f))$ allows us to assign a cost to cancer that remains at the end of the evaluation window, but we lack a reasonable *a-priori* estimate of the associated long term cost: for example, the cost associated with a small residual cancer population will depend strongly on whether it will rebound without treatment. For this reason we set $\phi(\mathbf{x}(t_f)) = 0$, and instead use extended time windows and evaluate the infinite horizon optimum solutions to check whether each optimal control calculated is free of finite time window artefacts, and reaches either a stable cancer free state or an optimal maintainance treatment regime within the time window (Sect D in S2 Text).

Note that the control term of the cost function is intended to include the health impacts of drug toxicity. As our model includes the off-target effects of drug treatment, the control term is directly linked with the death of healthy cells and so captures health impacts in this sense, although its generic form permits broader interpretations that also incorporate other side-effects of drug treatment. A plausible alternative that we did not adopt is to explicitly describe the health impacts due to loss of healthy cells, for example by including an additional term based on the gap between A and an assumed healthy level.

## Results

### Drug escape mechanism produces expected resistance to control

Our model extends the core features of the Sharp model [31], which we reproduce as the Null-N model by taking our full model and suppressing the drug resistant CD38- cancer cell population (N) and the off-target drug effect. We use this model as a negative control to validate the drug escape and off-target mortality effects. Simulations using the Null-N model are shown in Fig 2A–2D. Without treatment, the healthy and cancerous cells reach a balance. The presence of the immune response shifts this balance against the cancer without achieving elimination. But if the treatment can reduce the cancer level sufficiently, the immune response will prevent recurrence.

In Fig 2E–2H we show the results of the corresponding simulations using our full model. As intended, in the absence of the drug control the CD38- population plays only a marginal role; this can be seen in Fig 2E, 2F. In the presence of a high dose drug control (Fig 2G), the role of this population grows and has the effect of diminishing drug efficacy via the escape

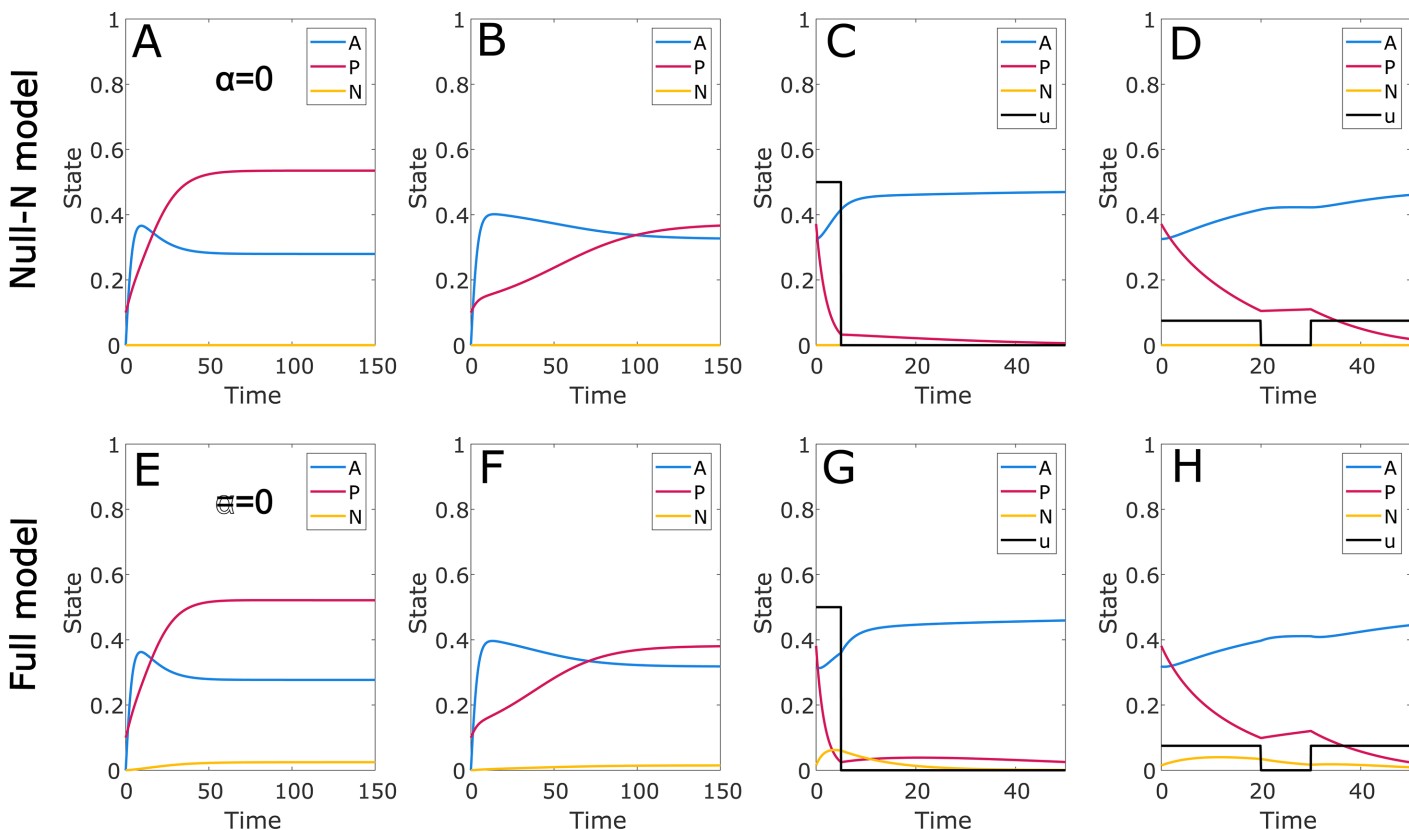

**Fig 2. Model validation and comparison with Null-N model.** Selected numerical simulations using the fourth order Runge-Kutta method and time step 0.001. The state variables representing the three cell populations (healthy cells, A; CD38+ cancer cells, P; and CD38- cancer cells, N) are each expressed as a proportion of the total theoretical carrying capacity. The full model developed in this paper (E–H) is compared to a simplified version without drug resistance mechanisms and off-target effects (Null-N model, A–D), in which the CD38- cancer population is suppressed. In panels A,B,E,F the initial state is $P = 0.1$, $A = N = 0$ and no control is applied; in panels A,E we also suppress the immune response. In C,D,G,H the simulation starts at steady state and a prespecified control $u$ is applied.

mechanism. Fig 2H, however, shows little or no difference in drug efficacy from Fig 2D, suggesting that the drug resistance mechanism plays a larger role when the control has higher intensity and shorter duration; we examine this issue more systematically below through the lens of optimal control. For the parameters used here, where the drug's effect on healthy cells is only one tenth of its effect on CD38+ cells, the off-target drug effect is extremely minor. However, in later results we consider higher values and see important effects.

## Drug resistance drives prolonged treatment

Using the linear and quadratic cost functions, we can evaluate the overall effect of the model modifications under the default parameters in terms of the cost to treat and optimal pattern of treatment (Fig 3). Note that cost values are not comparable between the two functions. In both cases, the total control and overall cost is increased relative to the Null-N control, and the duration of treatment is extended; the prolongation of treatment is less clear for the quadratic cost control due to its tapering form, but note that a smaller proportion of the total control is applied in the first 5 or 20 time units for the full model. For both cost functions a high initial drug dose in the full model rapidly reduces overall cancer levels, but produces much higher levels of the drug-immune CD38- population, and recovery of the healthy cell

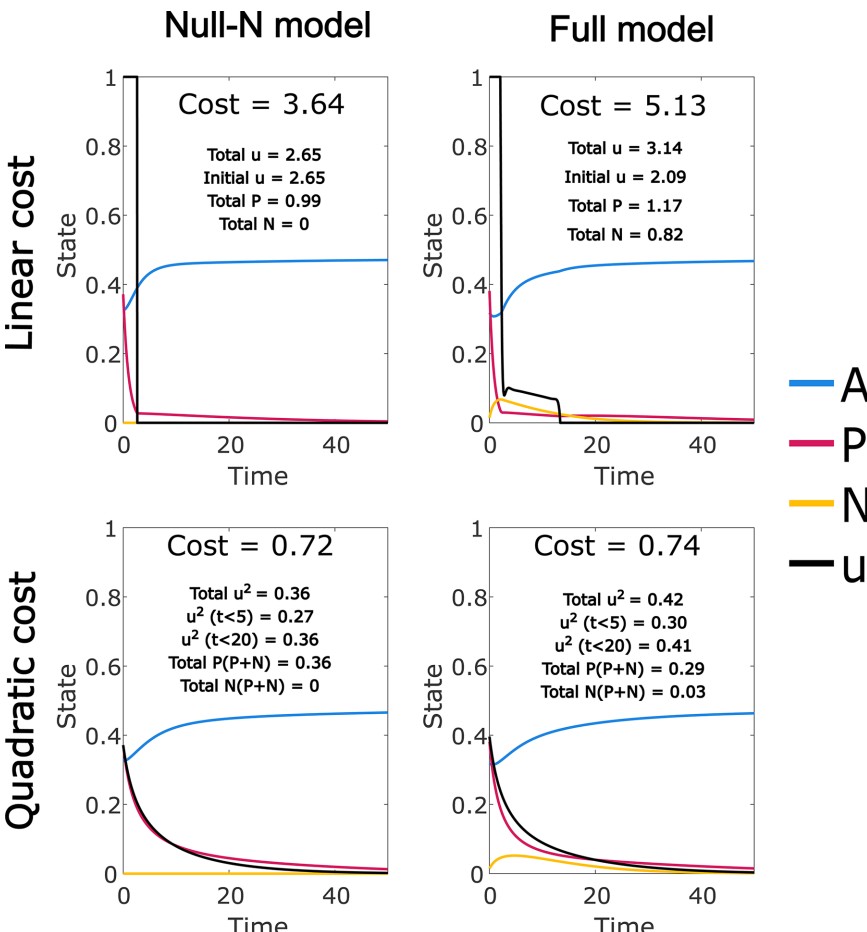

**Fig 3. Optimal controls for the full model developed in this paper (Full model) and a simplified version with drug resistance mechanisms and the off-target effect removed (Null-N).** In each case we have annotated the plot with the overall cost function value ((7) or (8)) and its components (control cost and cancer burden). We also note the total control applied in the initial period when the control is at its maximum level. For the continuous control cost function, the total cancer cost $((P + N)^2)$ is allocated proportionately between $P$ and $N$ for the quoted numbers. The drug related cost incurred in the first 5 and 20 time units is also given as an indication of relative control duration. Each of these totals represents the integral of the respective function over the interval $t \in [0, T]$, where $T = 50$ unless specified otherwise. Optimal controls were found using a time period of length 200, and results are shown for the initial 50.

population is inhibited by the off-target effect while the control dose is high. The control is then continued at a lower level that balances these factors, until immune suppression of the cancer is achieved.

The increases in cost, total control, and duration of treatment relative to the Null-N control are all much smaller for the quadratic than the linear cost. This is likely because the quadratic function (8) favours prolonged, low-level treatment, with the same total dose administered over a longer period incurring a lower cost. This explains the tapered shape of the quadratic cost control solutions for the Null-N model, and with the addition of the new mechanisms the required increase in treatment duration is small and achieved at low cost. However, if the true costs include any linear component such as the financial cost of drug supply, the quadratic

cost function will be least accurate at lower levels of control and cancer. Hence the very low cost associated with an extended period of low-level treatment may not be realistic.

In contrast, the linear cost function (7) does not discount the cost of continuing lower level treatment. Linear costs typically result in optimal control levels at either the maximum level or zero. In the Null-N model all solutions consisted of an initial period at maximum level followed by an abrupt and final end of treatment. This is known as a *bang-bang* control, although this term may refer to controls that are explicitly constrained to either a maximum or minimal value at each time ([34,35], or more recent work such as [36]). In our model, however, the linear cost function produces an initial period at maximum level followed by a period of intermediate level control. This intermediate period is known as a *singular arc*, and the control is a concatenation of bang and singular controls which we refer to as *bang-singular* [32]. The presence of this singular arc despite the linear cost function provides clear support for extended treatment.

The bang-singular control solution includes a small oscillation at the start of the singular arc, going briefly below and then above trend. This feature is invariant under increased numerical resolution, but additional iterations of the algorithm resulted in a very slow contraction of the oscillation in tandem with gradual steepening of the vertical drop immediately prior. We conclude that the oscillation is a local numerical artefact associated with imperfect convergence to large step changes in the control, and should be discounted. This does not apply to the general form of the control, or to more substantial breaks in control solutions.

## Reduced immune response requires prolonged treatment regime

The immune response, controlled by parameter $\alpha$, plays an important role in treatment. In Fig 4 we show the effect of varying this parameter on the optimal control. We see that a reduced immune response increases the cost to treat primarily through prolongation of the control; for the linear cost, the duration of the initial maximal treatment is almost invariant. As the immune response approaches zero, there is a point at which immune suppression becomes impossible at any cancer level, and the optimal control transitions to an initial high dose treatment followed by an indefinite maintenance treatment; this is associated with the majority of the cancer cell population becoming CD38 negative. We can also project ongoing costs from the trendline (see Sect D in S2 Text).

We are most interested in model parameters that allow both a persistent cancer state and the possibility of permanent control via drug treatment. We show in Sect E in S1 Text that this requires the Michaelis-Menten immune response; a linear immune response can be regarded as a simple modification of the exit rate parameters and cannot achieve the same effect. However, cases in which the cancer must be managed through ongoing treatment are also of interest, despite posing some difficulty in interpretation due to the finite time window used in the optimal control methodology. From a modelling perspective, it is significant that when the immune response is not sufficient to allow permanent control of the cancer, our algorithm is able to find an optimal steady state treatment regime, as shown in Fig 4 when $\alpha = 0$. This can be attributed to the additional mechanisms in our model, as it is not the case for the Null-N model. If we consider the Null-N model with no immune response and a constant level of control applied so that $P$ approaches 0, then $A$ will approach a steady state $A_0$ and we have $\frac{dP}{dt} \approx P(\rho_P(1 - A_0) - \mu_P - \mu_{Pu}u)$, giving an asymptotically exponential solution for $P$. Cessation of the control will lead to exponential increase until $P$ is again non-negligible. This implicitly models cancer at arbitrarily low levels, potentially less than a single cell, and also results in convergence failure of the optimal control, due to extreme insensitivity to control

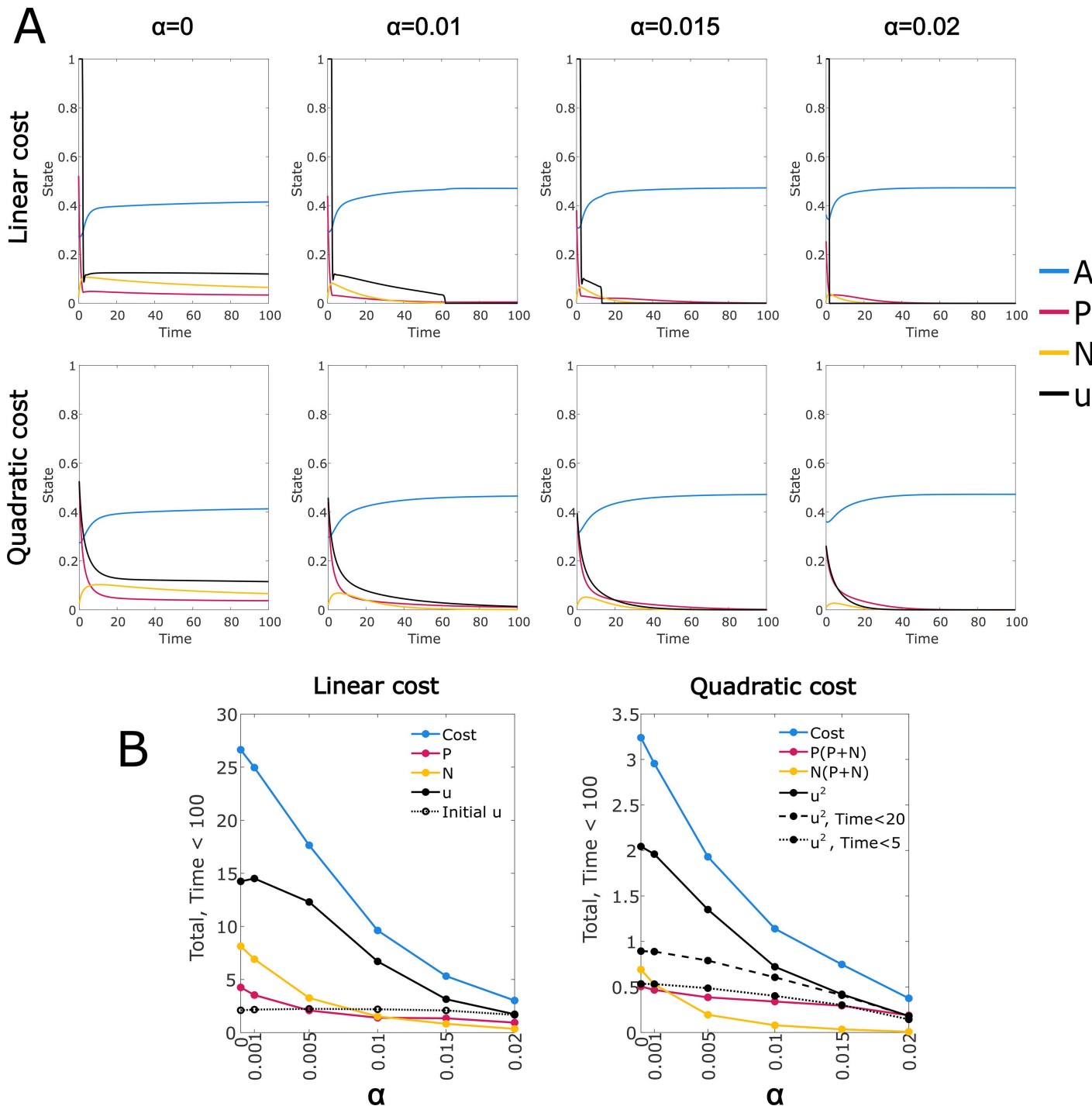

**Fig 4. Optimal control solutions for a range of values of α, which controls the strength of the immune response; default value is α = 0.015.** Drug duration is reduced for both cost functions as immunity is strengthened (left to right). A: Linear and quadratic cost optimal control solutions for selected values of α. B: Total costs and components up to Time=100. Each value plotted is the integral of the specified function over the interval Time ∈ [0, T], where T = 100 unless otherwise specified. For the "Initial U" value, T is the maximum value for which U = 1 over the interval [0, T].

timing while $P$ is at a negligible level. In contrast, cancer populations in the steady state treatment regime in Fig 4 remain non-negligible, and recurrence does not depend on exponential increase from an arbitrarily low level. Thus we see that our modified model provides an improvement in this case both computationally and as a biologically realistic system.

## Elevated off-target drug effect produces distinct form of linear cost control

In addition to modelling drug resistance via loss of CD38 expression, our model includes an off-target effect, in which the control causes some mortality in healthy cells. While the harm caused by drug side-effects can be modelled through the cost function, including this feature explicitly allows us to examine the effects on the population dynamics, and particularly the interaction with the drug escape mechanism. Since myeloma cells notably over-express the drug target CD38, we expect realistic values of the off-target mortality parameter $\mu_{Au}$ to be much less than 1. At this level the off-target effect appears to have limited influence. When we consider higher values (Fig 5) we see a general pattern of slightly increased costs (both drug and disease burden). However, at $\mu_{Au} = 1$ we see a striking change in the form of the optimal linear-cost control. Instead of a period of continuing control at reduced intensity (bang-singular), the initial period of maximum intensity control is followed by a complete cessation of treatment, then a second shorter period of maximum intensity treatment. During the break in treatment the healthy cell population recovers while the drug resistant CD38- population declines, but the CD38+ cancer subpopulation also recovers from low levels. The follow-up treatment prevents a cancer resurgence, reducing levels to where they can be suppressed by the immune response.

## Linear cost optimal controls may be cyclic or discontinuous

The linear cost optimal control solution with $\mu_{Au} = 1$ raises the question of whether the solution may take other forms depending on the choice of parameters, particularly cyclic or discontinuous control solutions. The value $\mu_{Au} = 1$ seems biologically implausible, so we performed a systematic search for alternate forms of the optimal linear cost control using a somewhat more reasonable value $\mu_{Au} = 0.5$. In Sect A in S2 Text we include a supplementary analysis of sensitivity to the parameters controlling the drug escape mechanism, using the approach shown above for the immune response and off-target effect. We note that the rate of expression switching appears to influence the shape of the linear cost control, while the drug-induced loss of CD38 expression has a strong influence on the prolongation of treatment, so we considered variations in these parameters as well as a reduced immune response.

When expression switching and drug-induced loss of expression are both slower than initially modeled (10-fold decreases in $\delta_P$, $\delta_N$ and $\delta_{P_u}$), a remarkable solution forms when the immune response is insufficiently strong (Fig 6, top row). The optimal control takes the form of short periods of control at maximum intensity, separated by longer periods of zero control. The initial treatment period is longer, and is also followed by a longer break; the treatment periods then follow a regular pattern. When the immune response is removed ($\alpha = 0$), permanent control of the cancer is not possible, and the solution tends towards a repeating cyclic pattern. When $\alpha = 0.01$ we see a modified version of this pattern which terminates when suppression by the immune response has been established. Increasing the rate of drug-induced loss of CD38 expression appears to suppress this cyclic solution (Fig 6, bottom row), however we retain a period of zero control following the initial period of maximum-intensity control. Note that the control for $\alpha = 0.01$ and $\delta_{P_u} = 2$ terminates at $t \approx 120$ (Fig G in S2 Text). The cases with unchanged or increased rates of expression switching ($\delta_P$ and $\delta_N$) did not give any cyclic or discontinuous solutions.

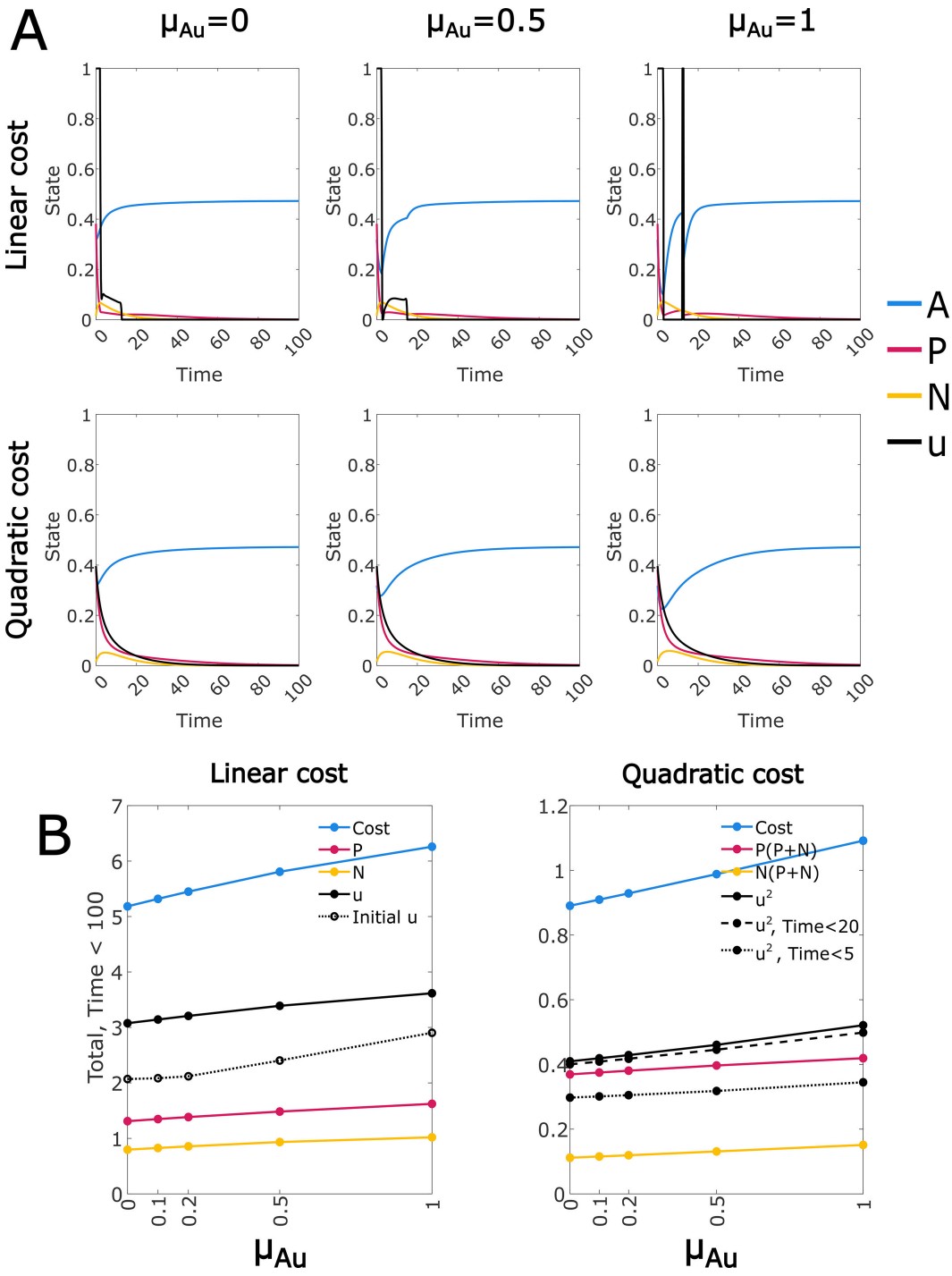

**Fig 5. Optimal control solutions for different off-target drug effect magnitudes ($\mu_{Au}$); default value $\mu_{Au}$ = 0.1.** Higher levels modestly increase costs but lead to a distinct change in the form of the optimal control for the linear cost function. A: Linear and quadratic cost optimal control solutions, selected values of $\mu_{Au}$. B: Total costs and components up to Time=100. Each value plotted is the integral of the specified function over the interval Time $\in [0, T]$, where $T$ = 100 unless otherwise specified.

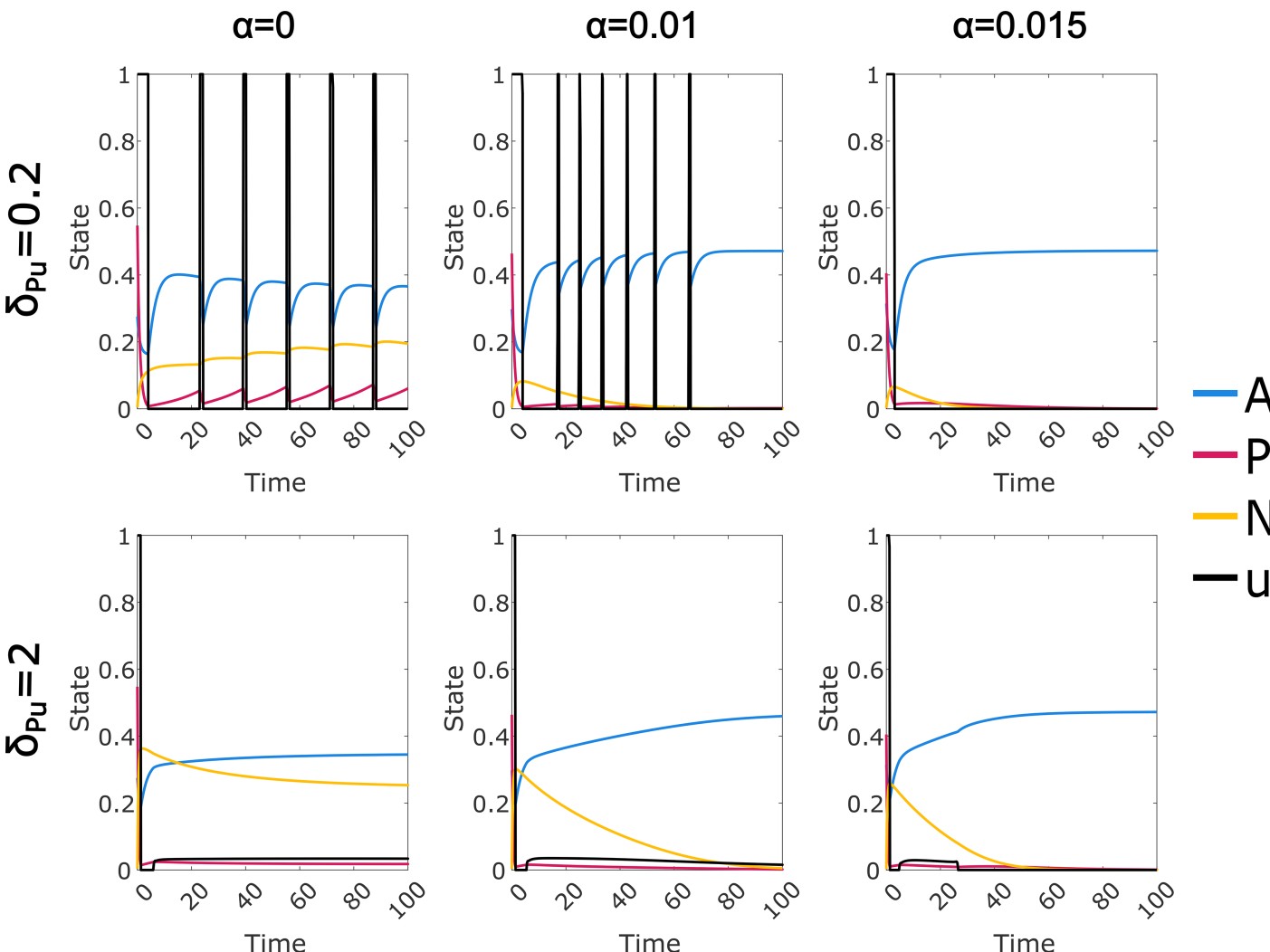

**Fig 6. Optimal control solutions for the linear cost function with increased off-target drug effect** ($\mu_{Au} = 0.5$; **default** 0.1), **reduced expression switching** $((\delta_P, \delta_N) = (0.0003, 0.003)$, **default** $(0.003, 0.03))$, **reduced or default immune strength** ($\alpha = 0, 0.01, 0.015$, **default** 0.15), **and default or increased drug induced loss of expression** ($\delta_{Pu} = 0.2, 2$, **default** 0.2). We observe several distinct forms of the control. In the two cases with $\alpha = 0.01$, the control was computed on an extended time window ($t \in [0, 400]$; see Fig G and discussion in Sect D in S2 Text).

The timing of the continuing cyclic form (Fig 6, top left) will be subject to a finite time window effect, although this has only a marginal effect on the solution (see Sect D in S2 Text). In addition, the two cases with $\alpha = 0.01$ were calculated on an extended interval ($t \in [0, 400]$). The reduced immune response, while still sufficient to prevent cancer resurgence, allowed a small cancer population to persist for a significant period; to avoid finite time window arte-facts it is necessary to ensure that cancer has been reduced to a negligible level by the end of the computational time window.

Since we see the cyclic solutions (Fig 6) at the lowest values of $\delta_P$, $\delta_N$ and $\delta_{Pu}$ that were considered in this experiment, it is natural to ask whether the complete removal of one or both of these features would also give optimal control solutions with a cyclic form. We retain $\mu_{Au} = 0.5$ and set $\alpha = 0$, consistent with the clearest examples of cyclic solutions seen. We observe (Fig 7) that the cyclic solution form appears to be consistent with $\delta_{Pu} = 0$, but not

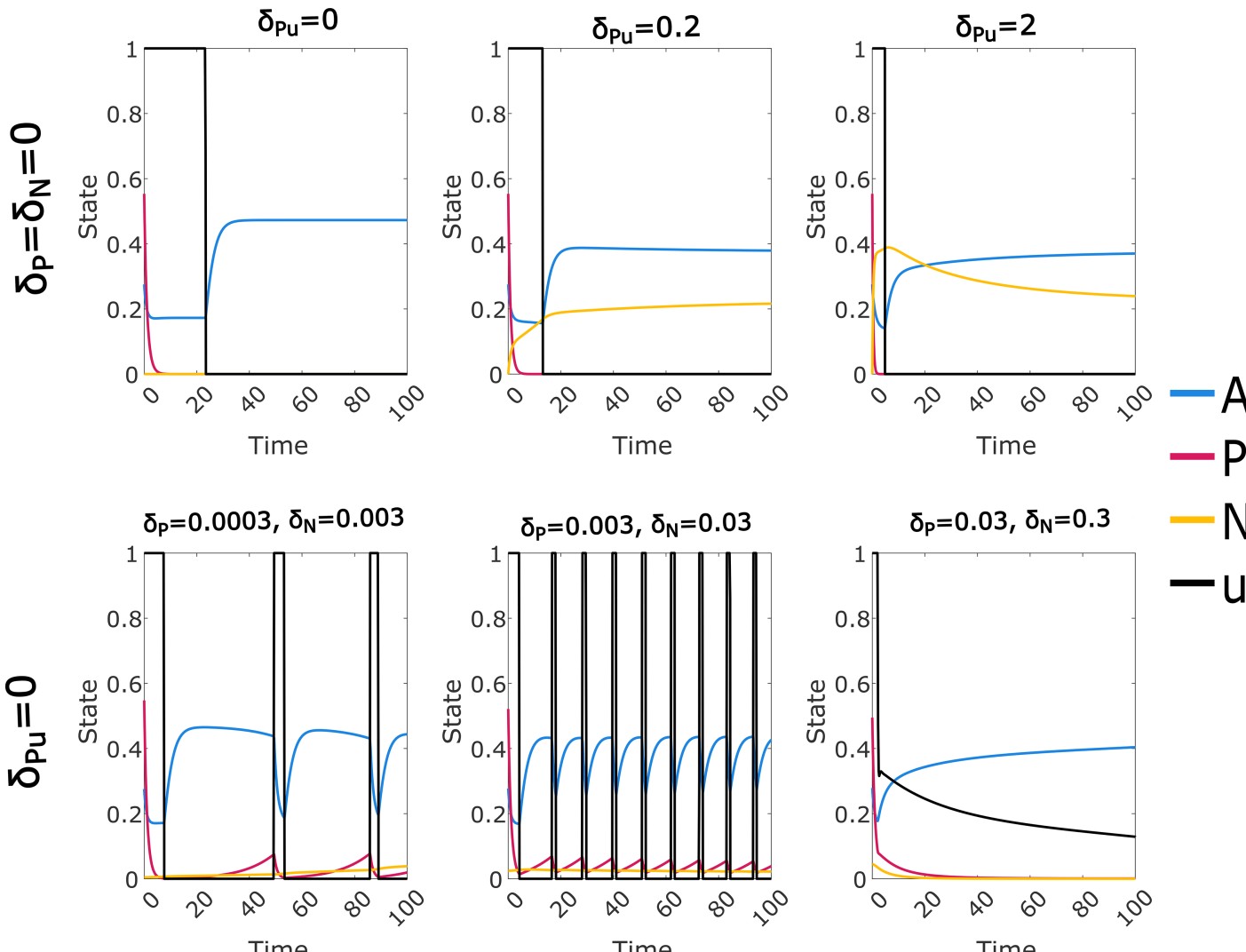

**Fig 7. Selected optimal control solutions for the linear cost function where either expression switching (top row, $\delta_P = \delta_N = 0$) or drug induced loss of expression (bottom row, $\delta_{Pu} = 0$) is removed from the model.** Other parameters are selected to be consistent with the cyclic solution forms shown in Fig 6; in all cases the immune response is set to zero ($\alpha = 0$) and the off–target drug effect $\mu_{Au}$ is set to 0.5 (default 0.1). We see that cyclic control solutions do not require the drug induced loss of expression mechanism, but do appear to require some level of expression switching.

with $\delta_P = \delta_N = 0$. In fact, the solutions with $\delta_P = \delta_N = 0$ appear to suggest that permanent control of the CD38+ cancer has been achieved, which is inconsistent with $\alpha = 0$; there is also an untreatable CD38- cancer population when $\delta_{Pu} > 0$. If we consider the nature of the model with these parameters, we observe that with no path from $N$ to $P$ and no immune response, we recapitulate the biologically implausible dynamics of the Null-N model with no immune response discussed previously. In this case we cannot expect convergence to an optimal steady state treatment regime; see also Sect D in S2 Text.

We have used a high value of the off-target mortality parameter $\mu_{Au}$ under the assumption that this is required to produce the cyclic solution form. We check this assumption by setting $\mu_{Au} = 0$ and $\mu_{Au} = 0.1$ (default value) in two cases with the clearest observed cyclic behaviour (Fig 8), leading to a loss of the cyclic form. Note that for the $\mu_{Au} = 0$ cases the convergence to

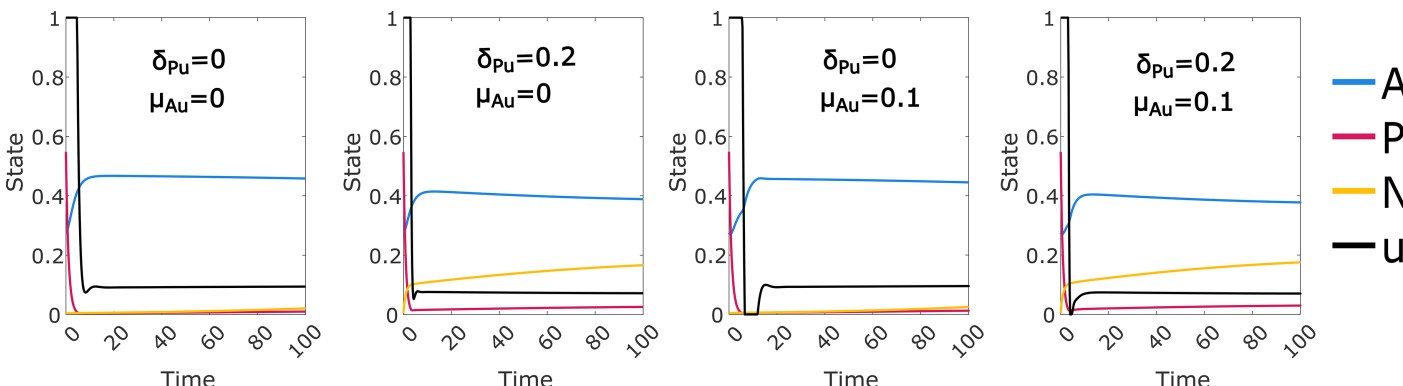

**Fig 8. Optimal control solutions for the linear cost function with zero or default off-target drug effect ($\mu_{Au} = 0, 0.1$), no immune response ($\alpha = 0$), reduced expression switching (($\delta_P, \delta_N$) = (0.0003, 0.003), default (0.003, 0.03)), and induced loss of expression set to either zero or the default value ($\delta_{Pu} = 0, 0.2$).** Parameter combinations which produced strongly cyclic optimal controls when $\mu_{Au} = 0.5$ no longer do so when $\mu_{Au}$ is reduced.

steady state treatment occurs over a much longer time period than shown, with a very gradual increase in $N$ (Table A in S2 Text).

While this analysis does not amount to a full exploration of the parameter space, our investigation suggests that the cyclic form requires a high value of $\mu_{Au}$, a low value of $\alpha$, and a low but non-zero value of $\delta_N$ and $\delta_P$.

## Discussion

We have presented a model of myeloma treatment using the monoclonal antibody Dara, with which we investigated the impact of a drug resistance mechanism and off-target cell mortality using optimal control theory. In our model myeloma cells resist the effect of Dara via loss of CD38 expression, albeit at the cost of reduced fitness. This loss of expression may result from either differential mortality or as a direct result of drug exposure. The proposed mechanisms generally resulted in increased overall costs and extended duration of treatment.

The Null-N model, which we used as a negative control without the drug resistance and off-target mechanisms, gave optimal control solutions of two forms, depending on the cost function. The linear cost function with bounded control values gave solutions of the common "bang-bang" form, in which the control started at the maximum level and then at some time point permanently switched to zero. Intuitively, there is no advantage from delay in treatment, and so the total drug dose necessary to contain the cancer is administered in the minimal possible time. The quadratic cost function gave continuous solutions which began at a high level then dropped continuously, a trade-off between removing cancer as quickly as possible and the cost advantage of treatment at lower dose.

When we included the drug resistance and off-target effect mechanisms in the model, we found a clear increase in the overall cost and the treatment duration under both costs functions (Fig 2). The effect was smaller for the quadratic cost function, with no apparent change in the control form. We attributed this to the reduced cost assigned to extended low intensity treatment. For the linear cost, however, the optimal control solution had a "bang-singular" optimal control form, in which the initial period of maximum level control was followed by an immediate reduction and a period of lower-intensity treatment, decreasing linearly before stopping abruptly. We can understand this as a period in which the imperative to treat the cancer must be balanced against the need to allow time for the CD38 expression level in the

myeloma cells to recover, so that the drug effectiveness is restored. Recovery of the healthy cell population may also be a factor in this pattern.

We analysed variation of the parameters controlling the drug resistance and off-target effect mechanisms, as well as the immune response (Results and Sects A, B in S2 Text). In most cases the relationship between the rate parameters and outcomes such as total drug dose, treatment duration and cancer persistence were found to be at least directionally consistent between the two cost functions considered, suggesting that the identified trends are robust. Exceptions included the rate of expression switching (Fig A in S2 Text), which had very little effect on the quadratic cost control, and the rate of drug-induced loss of CD38 expression, which had a somewhat inconsistent effect (Fig C in S2 Text). Increases in expression switching, drug-induced loss of expression, and CD38- fitness all induced increased duration of control under the linear cost, with somewhat complex effects on the control shape. However, the clearest prolongation of the control was induced by a reduced immune response, with a one third reduction causing a large effect under both cost functions (Fig 4). Complete removal of the immune response resulted in controls that converged to an optimal steady state treatment regime, provided that expression switching was not removed.

The quadratic cost function results helped to demonstrate the sensitivity of the control form to the cost assumptions, as well as the robustness of many of the trends we identified. However, we conclude that these controls are of less interest than the linear cost controls, notwithstanding the greater ease of computation. Indeed, the literature provides little support for a super-linear health burden of either cancer [37] or Dara [38,39]. Even if super-linear costs do exist, the very low cost assigned to an extended period of low dose treatment is unlikely to be realistic; if real costs include at least some linear factors, such as the financial drug cost, these will tend to dominate as $u$ and $P + N$ approach zero. This unrealistic low cost gives rise to the tapering form of the quadratic cost controls, and appears to substantially obscure the effect of the drug resistance mechanism, particularly in terms of the prolongation of treatment. In contrast, for the linear cost function any period of ongoing control at a reduced level can be reasonably attributed to the biological mechanisms that we model.

The most notable effect on the shape of the linear cost optimal control was caused by an increase in the off-target drug effect (Fig 5). When combined with reduced but non-zero CD38 expression switching and reduced or zero immune response we obtained controls of a cyclic bang-bang form, with an initial period of maximum control followed by a gap then repeated smaller periods of maximum treatment, continuing either indefinitely or until immune suppression was established (Fig 6). We are not aware of optimal controls of this form being identified previously in a cancer model, although bang-bang controls with a delayed start (off, on, off) have been found for a cancer immunotherapy model [27]. Drug induced loss of expression ($\delta_{Pu}$), despite prolonging the control, appears to suppress the cyclic pattern. Increased $\delta_{Pu}$ did give a additional optimal control form, bang-singular with a clear gap before the singular arc, which could be either finite or indefinite.

Our study is largely theoretical, using parameters that have not been quantitatively fit to data. Indeed, mechanisms such as those in [13] are unquantified; the natural variation of CD38 expression between myeloma cells, and the rate at which this changes naturally and in the presence of Dara, is unknown. As a result, our optimal control results do not serve as clinical recommendations. Rather, they highlight how the different aspects of the disease are expected to influence how the disease best be treated. More generally, we demonstrated the value of the controls in identifying which mechanisms within a complex model affect the model's optimal control. In clinical practice, Dara treatment consists of discrete multi-hour infusions, typically repeated at weekly or longer intervals, with multiple dosing

levels used [40]. Any attempt to calibrate our model to real data and draw a closer connection to clinical practice would have to account for this intrinsically intermittent process (as well as pharmacokinetics); optimal controls containing extended periods of low dose treatment would need to be interpreted in this context. Furthermore, a fully realistic cost function for a diffusion treatment such as Dara would likely incorporate a fixed per-session cost. This cannot be directly represented in the form of (3), and finding a method of incorporating such a cost factor could be of value. Nevertheless, our demonstrated capacity to identify the model dynamics that favour extended treatment at reduced intensity, or more continuous vs discrete treatment, has potential to inform treatment regimes. In contrast, the quadratic cost controls we obtained lacked this qualitative information, while linear cost controls of an enforced bang-bang form would be less informative given the availability of multiple Dara dosing levels.

We modelled the loss of CD38 expression in myeloma as a temporary process that will fully reverse if treatment stops, which is consistent with clinical observation. In fact, permanent failure of Dara appears to be linked to an increase in expression of the CD38 complement inhibitors CD55 and CD59 [41]. An interesting extension to our model would be to include a permanently resistant population which increases in response to treatment. Optimal drug treatment in the face of such acquired resistance is an important area of ongoing research. Another critical context is microbial immunity; see for example [42], which optimises antibiotic therapy using reinforcement learning, a machine learning method which has shown promise as an alternative to traditional optimal control theory. Reinforcement learning has also been used in recent research on cancer therapy [43].

The more complex linear cost optimal controls posed several challenges for the numerical methods used. As discussed in Sect C in S1 Text, the initial algorithm adopted for the linear cost function required minor modification to obtain convergence for bang-singular controls. Moreover, this convergence was imperfect at discontinuities, with local artefacts at bang-singular interfaces. The interpretation of results from a finite time window method in an intrinsically infinite-horizon context also demands careful consideration. In most cases we were able to demonstrate that the controls converged (in the time variable, not iterations) to an infinite-horizon, optimal time-averaged cost constant control; most often this was complete elimination of the cancer (Sect D in S2 Text). An exception was the indefinitely repeating bang-bang solution form, for which the window length appeared to bias the cycle timing. Aside from these two specific issues, our sensitivity analyses indicated that the control forms we presented are robust; for example, in Sect C in S2 Text we saw that for a composite but mostly linear cost function, the control obtained using the continuous control algorithm approximated the bang-singular form. Therefore, we are content to present the results with the caveats noted above. However, the calculation of bang-singular controls is a subject of current research (for example [44,45]), and the artefacts we saw point to the value of these more complex, multi-stage methods in order to efficiently find these controls with full fidelity. A further methodological refinement would be required to deal with the finite window bias on the indefinitely repeating bang-bang control. Finally, for either cost function the method used does not exclude the risk that the optimal controls are only locally optimal, and the result could in theory depend on our initial control choice $u(t) = 0$ for all $t$. The generally intuitive and explainable shape of the controls suggests that this risk was minimal; for example, an initial delay in treatment would not provide a benefit. Nevertheless, we must acknowledge this limitation.

We briefly note some further limitations of this study. Firstly, we emphasize that our modelling effort is quite theoretical, focused on a particular mechanism of drug resistance and its consequences for optimal drug treatment. Although we believe that factors such as a space

constraint and immune response are relevant to multiple myeloma, the mechanisms adopted from the Crowell model [30] are not adapted to the specific biology of multiple myeloma. The core drug resistance mechanisms were also simplified and lack calibration to real data. The use of a stable equilibrium as the initial state when calculating optimal controls, adopted from [31], provided a consistent basis for comparison, and can be regarded as a theoretical worse case in terms of the difficulty of gaining control. A more realistic initial state would be based on a selected stage in a calibrated model of cancer progression. Finally, we note that we did not attempt to definitively characterise the optimal control solutions across the entire plausible parameter space. This is largely due to the complexity of the system, but the reduction in convergence speed for the more complex linear cost controls was also a limitation, and improvements to the convergence algorithm could allow a more complete analysis.

## Supporting information

**S1 Text. Supplementary methods.** Detailed numerical methods used, including required mathematical derivations. This complements the methods described in the main text.
(PDF)

**S2 Text. Supplementary results.** Several sensitivity and other additional analyses.
(PDF)

## Author contributions

**Conceptualization:** James Lefevre, Brodie A.J. Lawson, Pamela M. Burrage, Diane M. Donovan, Kevin Burrage.

**Formal analysis:** James Lefevre.

**Funding acquisition:** Diane M. Donovan, Kevin Burrage.

**Methodology:** James Lefevre, Brodie A.J. Lawson.

**Software:** James Lefevre.

**Supervision:** Diane M. Donovan, Kevin Burrage.

**Visualization:** James Lefevre.

**Writing – original draft:** James Lefevre.

**Writing – review & editing:** Brodie A.J. Lawson, Pamela M. Burrage, Diane M. Donovan, Kevin Burrage.

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
