## [Decision Letter · Decision Letter 0]

7 Oct 2024

Dear Dr Lefevre,

Thank you very much for submitting your manuscript "Optimal control of Multiple Myeloma assuming drug evasion

and off-target effects" for consideration at PLOS Computational Biology.

As with all papers reviewed by the journal, your manuscript was reviewed by members of the editorial board and by several independent reviewers. In light of the reviews (below this email), we would like to invite the resubmission of a significantly-revised version that takes into account the reviewers' comments.

Three reviewers have now weighed in thoughtfully on this manuscript, and have come to very similar conclusions and concerns. While I am suggesting major revisions, I'd like to emphasize that the effort to improve the paper that the reviewers have expended is significant, and the revision must therefore be. In particular, the shared concern that this is purely theoretical model, and that the authors should make some attempt to contextualize their model (parameters) in either some inferred set from clinical studies, or potentially from experimental work.

There also seems to be some shared concerns about the use of specific terms/definitions from optimal control theory. There remains within the work, however, meaningful novelty that was appreciated by all reviewers, for which the authors should be commended, and on which the decision to recommend revisions strongly rests.

We cannot make any decision about publication until we have seen the revised manuscript and your response to the reviewers' comments. Your revised manuscript is also likely to be sent to reviewers for further evaluation.

Sincerely,

Jacob Scott, MD

Academic Editor

PLOS Computational Biology

Pedro Mendes

Section Editor

PLOS Computational Biology

Three reviewers have now weighed in thoughtfully on this manuscript, and have come to very similar conclusions and concerns. While I am suggesting major revisions, I'd like to emphasize that the effort to improve the paper that the reviewers have expended is significant, and the revision must therefore be. In particular, the shared concern that this is purely theoretical model, and that the authors should make some attempt to contextualize their model (parameters) in either some inferred set from clinical studies, or potentially from experimental work.

There also seems to be some shared concerns about the use of specific terms/definitions from optimal control theory. There remains within the work, however, meaningful novelty that was appreciated by all reviewers, for which the authors should be commended, and on which the decision to recommend revisions strongly rests.

Reviewer's Responses to Questions

**Comments to the Authors:**

Reviewer #1: Review is uploaded as attachment, for ease of access to hyperlinks in the review. I have duplicated the summary section below.

Summary (Duplicated from attached document) =================

The authors proposed to use optimal control techniques to derive treatment schedules, within the disease context of multiple myeloma. This was conducted on an ODE model of a partially drug resistant tumour, building on previous work by Khalili et al. and Sharp et al. (who are cited in the manuscript). Novel to this paper, the authors also allowed for both drug evasion (via reversible CD38 expression) and off-target killing of healthy cells.

The inclusion of these additional phenomena is well motivated within the study. It is worth noting that the study is purely theoretical (neither the ODE model nor the derived treatment regimens are parameterised by nor compared to experimental/clinical data), however the authors do well to convey those limitations. The authors also provide code for their model, although I have detailed difficulties in running this below.

These additional drug dynamics motivate quantitative (i.e. extended treatment times in Sections 2.2 & 2.3) and qualitative changes (Sections 2.5 & 2.6) to the optimal treatment approach. The latter are particularly exciting, demonstrating that optimal binary dosing strategies may consist of either single or multiple treatment periods depending on the parameter regime. Within the context of this class of models, this appears to be the first demonstration of optimal cyclic controls.

However, the authors employ an unconventional form of optimal control, that I believe is mis-categorised in the paper as ‘bang-bang’ control; I provide more details of prior approaches to cancer therapeutic scheduling using conventional ‘bang-bang’ control below. I recommend that the authors reframe the narrative of this paper to focus on the impact of different objective cost functions (within a continuous control framework). This would build upon established literature in the field, and I believe would also present their current results through a stronger narrative which could be of wider interest.

Reviewer #2: See attachment

Reviewer #3: Review of "Optimal control of Multiple Myeloma assuming drug evasion and off-target effects" by Lefevre et al.

The authors propose an ODE-based model for Multiple Myeloma (MM) plasma cell cancer treated by

a monoclonal antibody medication (Daratumumab), which targets the CD38 receptor usually over-expressed in myeloma cells.

The two main complications are

* that the CD38 expression might switch off in (some of) the cancer cells (resulting in a cancer subpopulation unaffected by the drug) and

* that many healthy cells also have a significant level of CD38 expression (resulting in negative side-effects for the patient).

The model also assumes a (constant rate) influx of upstream healthy-cells, a shared carrying capacity for the competing

healthy and cancer cells, and an immune response (modeled by Michaelis-Menten terms in the ODEs).

The authors then apply the tools of control theory (via Pontryagin Maximum Principle, PMP) to find the "optimal" rate of

administering Daratumumab. The optimality is defined over a fixed time horizon and with respect to two different

optimization criteria. A substantial part of the paper is devoted to analyzing the simulation results based on these

drug/control policies and comparing their qualitative behavior in response to parameter variations. The paper is concluded by

the Methods section, which provides a fairly detailed description of numerical methods and a stability analysis of no-drugs equilibria.

As an applied mathematician working in the same space, I value papers that advance the use of optimal control techniques

in cancer modeling. However, I have significant concerns/questions (detailed below) about some of the fundamental modeling

and optimization choices/assumptions made by the authors. I am also not sure whether the current style of the narrative is

a good match for the majority of PLOS CB target audience. I thus recommend major revisions, expecting that

the resubmitted version will be substantially different.

Major Concerns/Comments:

1) The style/logic of the narrative and the scarcity of biological details

The manuscript is mostly well-written, with a significant amount of details included to explain the mathematical model

and the drug policy optimization based on it. It is also great that the authors are candidly

discussing the computational challenges (e.g., associated with approximating optimal bang-singular controls), but

the logic of the narrative appears somewhat strange: the authors occasionally make it sound like some of

the modeling/optimization choices are made purely for mathematical & computational reasons. They often do not

provide sufficient information for readers to evaluate whether those modeling choices/assumptions are biologically

justifiable.

In general, the paper is very light on biological details. E.g., there is a separate section 1.1 on dynamical systems &

optimal control, which is written in a very general way. It includes 2 paragraphs on some ODE-based cancer

models, but provides no information on why prior models are not suitable to describe MM and Dara-based therapies.

After all, cancer modeling literature is vast and the emergence of drug-resistance + damage due to drug toxicity are

prominent feature in many papers -- so, what is different here? Why is the proposed new model an improvement?

Explaining this is particularly important since the model is at the heart of the paper while the control techniques

employed by the authors are mostly fairly standard.

2) Modeling assumptions

Since the presented model is meant for a theoretical study only, it would be unreasonable to ask for

a detailed justification of all biological mechanisms. But I think the readers would still expect to learn why

specific assumptions are reasonable (even if clear simplifications). Here are a few questions I was

left with:

* Why is there a shared carrying capacity for A, N, and P cells?

Does it represent some competition for resources or volume constraints?

If the latter, is it really compatible with a constant influx of healthy cells assumption?

Note that even in the absence of N & P, the number of A cells will equilibrate above 1.

* Why is it reasonable to assume that the immune response is independent of the current A abundance?

* The following paper develops a much more detailed model of the immune response to MM:

https://www.science.smith.edu/~ntania/papers/Gallaher_Springer_2018a.pdf

Is it compatible with the authors' basic assumptions?

* In many Figures (starting from 1c,d,g,h and throughout), the computational experiments start from

a (drug-free) steady state with the highest observed abundance of cancer cells.

Does such equilibrium actually exist for MM?

Is this a reasonable initial condition for treatment?

Wouldn't a patient die long before this equilibrium level is approached?

* Many of the tests seem to imply that, at least for a range of parameter values, the system is bi-stable,

with a cancer-free equilibrium (A_0, 0, 0) also stable, and the natural (drug-free) dynamics leading to it

once P is sufficiently suppressed by drugs.

Was this bi-stability confirmed at least numerically for the case alpha>0 ?

Did the authors try to identify the separatrix for the basins of attraction?

Is this bi-stability biologically justifiable?

Will the real MM population continue declining to zero _after_ the drug therapy is finished?

* The authors use the "general overexpression of CD38 in myeloma" to argue that

proliferation rate rho is higher while the natural death rate mu is lower for P cells compared to N cells

_and_

that the rate of P  N transitions should be much lower than the rate of N  P transitions.

Is there any biological evidence for both statements to hold true?

(Either of these assumptions can produce experimentally observed overexpression of CD38.

Why assume both?)

Fig.5 provides some experiments on the efficacy of optimal drug therapies when both

delta_P and delta_N are scaled down proportionally, keeping their ratio fixed.

If there is no biological reason to believe that the ratio of 10 is reasonable? Wouldn't it be prudent

to conduct some tests with smaller ratios? (In all included experiments, the abundance of N

cells remains fairly low with or without drugs. Does this correspond to what is observed about

MM in practice?)

Please address all of the above points in the paper -- your readers will be grateful.

3) Optimization: objectives, horizons, interpretations

* It seems strange that the running cost includes no penalty for the depletion of healthy cells (A).

The abundances of P & N are only a part of the story. The same "successful" therapy might be

prohibitive if starting from a lower A abundance. If the authors don't want to do this, another

alternative is to introduce a state constraint -- the minimum level of A allowed throughout the process.

If the authors decide against these changes, they should explain to readers why ignoring this aspect

is reasonable.

* The authors are considering a fixed-horizon optimal control problem with no terminal cost, which

can creates all kinds of strange "the deadline is near" effects (as they correctly point out at the end of

section 1.3). The authors handle this by assuming that the planning horizon is large (T=200) when

computing optimal controls, but then using those controls and plotting all trajectories for a much

shorter time interval (e.g., [0,T_1] with T_1 = 50 or 100). This is not a standard/robust approach

since one never knows a priori whether the T/T_1 ratio is large enough. At the very least, it is

necessary to recompute the optimal control for the horizon 2*T and check that on [0,T_1] it remains

virtually unchanged. A much better / theoretically-sound approach is to use a suitable terminal cost

$\phi(A,P,N)$, which should properly assign a higher penalty for N rather than P at the end of the process --

since a sizable CD38- population makes that cancer harder to control and the recurrence more likely.

The authors assertion that $\phi=0$ is "generally preferred as it provides more tractable computations"

seems very strange. PMP is routinely used with nonzero terminal cost in a variety of applications.

* The authors are at times discussing a possibility that "permanent control" is needed and of finding

an "optimal steady state treatment" for "indefinite maintenance" and of projecting "ongoing costs from

the trendline". All this is not really compatible with the fixed-horizon formulation -- the authors indirectly

acknowledge some of these difficulties (e.g.,on line 239), but their readers deserve an explicit and

detailed discussion of what they have in mind.

A rigorous case for the "steady state treatment" would require a different optimal control formulation

-- with infinite horizon and either time-discounting or time-averaged optimality.

* It is not clear whether u and u^2 used in their running costs are meant to reflect the monetary cost of

drugs only (which makes u^2 hard to justify) or also reflect the overall burden of treatment for the patient.

The discussion on lines 367-377 seems to provide two contradictory answers to that question,

additionally muddled by a sentence on lines 216-217 on p.11.

It seems that a more logical choice would be to use a more general superlinear cost; e.g.,

$u + \kappa u^2$, where a positive parameter $\kappa$ could be tuned as needed.

Note that repeating the computation with a sequence of decreasing $\kappa$ values could produce

a nice sanity check: the overall cost would tend to what can be found in the authors' discontinuous

control optimization (with the running cost L=u+P+N).

* For the first examined running cost (L=u+P+N), the authors are referring to the resulting control as "bang-bang"

throughout the paper. But this leads to a significant terminological confusion: this term is uniformly used in

the literature in situations where the optimal control takes extreme values only. Problems, in which

the Hamiltonian is linear in u but the switching function \psi might stay equal zero over some interval, yield

"singular control arcs" with $u \in (0,1)$ for that duration. Whenever singular arcs are present, the standard

nomenclature is "bang-singular" or "bang-singular-bang" controls -- precisely the setting for many examples

in the current paper (e.g., top right panel in Fig 3 and top row examples in many subsequent Figures).

The authors should switch to this standard terminology to avoid confusing their readers.

4) Optimization: numerical methods

* It is well-known that PMP is merely a necessary condition for a local optimality of control/trajectory pair.

[This is in contrast to the Hamilton-Jacobi-Bellman dynamic programming approach, which guarantees

the global optimality, but at a higher computational cost of solving a PDE.]

Without special properties of a control problem, one cannot guarantee more tan the local optimality

of a PMP-based solution. (And even this requires verifying a second-order Legendre-Clebsch condition.)

In general, a solution to the two-point BVP specified by the authors' equations (1,2,9,10) may not be unique.

Which among them will be recovered by a numerical method depends on the chosen initial guess for

the control u. Only one among those solutions will correspond to a globally optimal control/trajectory

and finding the right initial guess to converge to that global optimum can be quite non-trivial.

The authors should

- either explain why they believe this two-point BVP has a unique solution for their specific control problem

- or explicitly acknowledge that the obtained controls might be only locally optimal and there might well exist

another control that produces a lower overall cost.

* The authors' approach to computing bang-singular controls is somewhat ad-hoc and the resulting controls

likely contain numerical artifacts.

All of their graphs with bang-singular controls show an overshoot -- a brief sharp dip in the u value whenever

the singular arc starts, followed by a brief (much smaller) increase and a slower, seemingly linear-in-time

decrease in u. That initial downward overshoot looks strange. The authors should check whether it remains

unchanged or decreases/vanishes under numerical refinement -- i.e., when recomputed with much smaller

time-steps in their Runge-Kutta solver and with more stringent convergence criteria in their forward-backward

iterations.

If these overshoots are indeed numerical artifacts, this should be acknowledged explicitly in the paper.

It is also worth mentioning that bang-singular controls are an active research area and that there are a few

well-developed numerical methods for computing them accurately. See, for example,

https://doi.org/10.1007/s10589-022-00350-6

https://doi.org/10.1016/j.compchemeng.2020.106923

and references therein. The authors might find it worthwhile to switch to one of these instead.

Minor comments:

* the word "on" on line 54 (p.3.) should probably be deleted;

* the statement on lines 101-102 (p.5), just after formula (3), is only correct when f is also linear in u.

This is true for your example, but at this point in the paper you are still discussing a general control

problem with a general f.

* line 124, p.6: "cancer calls"  "cancer cells"

* p.7 It would be really helpful to provide some information on the hypothesized mechanisms of expression

switching. This will make it much easier to evaluate the suitability of terms in the ODE and the realistic

range of values for the relevant parameters.

* line 140: please define "fitness"

* p.8: The paper states that

"Where possible, parameter values were adapted from the Sharp model, which were selected

to produce balanced dynamics supporting both healthy and cancerous states and the capacity

for effective drug control."

Please provide a precise statement on which parameters have exactly the same values as in

the Sharp model.

* line 155: "population N substantially less fit"  "population N is substantially less fit"

* p.10 & beyond: Please introduce the "Null-N model" formally.

Please don't refer to it as "Sharp model" in figure captions.

* line 189 states that $\mu_u = 0$.

I suspect that you meant $\mu_{Au} = 0$.

* Section 2.2: please specify explicitly what is meant by a "prolonged treatment".

* Fig. 3 and beyond: the legends are confusing.

E.g., I suspect that when you write $U^2 (t<5)$, what is really meant is $\int_0^5 u^2(t) dt$, but I am not completely sure.

Please explain this explicitly in the text. The same issue is also relevant in subsequent figures (e.g., bottom right Fig.4).

* line 230: please define what is meant by the "final control of cancer".

* line 245: "asyptotically"  asymptotically"

* Most of the experiments in section 2 can be viewed as sanity check; i.e.,

they don't produce insights beyond what one would expect without computing optimal controls.

(E.g., yes, more drugs will be needed to control MM if the immune mechanism is less effective or

if drug-escapement is enabled. This seems quite natural.)

I would recommend devoting more space to discuss a few unobvious observations -- e.g., the non-monotonicity

of the overall cost as a function of drug-induced escapement under "continuous optimal" control

-- the bottom right in Fig.7.

* Section 2.6 is also interesting (might be worth drawing a connection to "metronomic" drug therapies), but

the chosen fixed-horizon optimization framework makes it harder to discuss these results rigorously.

Definitely worth checking if these approximately optimal policies change much under numerical refinement.

* While it's up to the authors to determine the structure/composition of their paper, I did not find the "Conclusions"

contributed much beyond what was already covered just above. I would suggest omitting section 3.1 altogether.

* The usual format in PLOS CB seems to indicate that section 4 would look more natural as Supplementary Materials.

* The discussion on lines 475-483 is hard to parse due to the choice of notation.

I would advise against using the symbol $u'(t)$.

Please also define formally what is meant by $u_t$.

* The authors provide a github repository link in the paper, but it appears to be invalid:

https://github.com/jameslefevre/optimal-control/tree/main

**Have the authors made all data and (if applicable) computational code underlying the findings in their manuscript fully available?**

Reviewer #1: Yes

Reviewer #2: Yes

Reviewer #3: **No: **The authors provide a github repository link in the paper, but it appears to be invalid:

https://github.com/jameslefevre/optimal-control/tree/main

PLOS authors have the option to publish the peer review history of their article (what does this mean?). If published, this will include your full peer review and any attached files.

Reviewer #1: No

Reviewer #2: **Yes: **Davis T Weaver

Reviewer #3: **Yes: **Alexander Vladimirsky
---

## [Decision Letter · Decision Letter 1]

23 Jun 2025

Dear Dr Lefevre,

We are pleased to inform you that your manuscript 'Optimal control of Multiple Myeloma assuming drug resistance

and off-target effects' has been provisionally accepted for publication in PLOS Computational Biology.

Best regards,

Jacob Scott, MD

Academic Editor

PLOS Computational Biology

Pedro Mendes

Section Editor

PLOS Computational Biology

Reviewer's Responses to Questions

**Comments to the Authors:**

Reviewer #1: The authors have responded carefully and comprehensively to my suggestions. In particular I appreciate the reworking of the presentation of the bang-singular approach, and the move of Section 2.4 into the supplementary material. I also wish to commend the authors on the inclusion of a supplementary section exploring the relative weighting of tumor size (i.e. the health cost of the cancer) and cumulative drug (i.e. the financial cost of treatment) in the cost function. The updated supplementary information is a strong addition to the paper, though authors could consider integrating the figures into the text where they are referenced for readability.

Overall the new manuscript, while lengthy (especially in the Discussion), is much clearer to read. My concerns around the presentation of ‘bang-bang’ control have been resolved, and while the mathematical model remains disconnected from biological or clinical data, the authors present a much more comprehensive evaluation of this limitation in the discussion. The updated manuscript presents a methodologically-sound and useful contribution to the wider literature on this topic.

Specific Comments:

I would like to commend the authors on their additional discussion around the mathematical form of the cost function. However I still have minor issues with their original motivation of quadratic costs - the authors say ‘Quadratic functions are also widely used, with the justification that super-linear costs are evident in many contexts.’ I cannot see any justification for the drug cost scaling super-linearly with the dosage. In practical terms, the financial cost of the drug scales linearly with the total dosage (or perhaps sub-linearly in some insurance-supported contexts). The scaling of experienced toxicity with total dosage is more complex, with much literature about sub/super linear dose response curves (e.g. https://doi.org/10.1371/journal.pcbi.0030024), but typically the toxicity is observed to scale sub-linearly with the cumulative dose (e.g. https://pmc.ncbi.nlm.nih.gov/articles/PMC6855968/ and comments in previous review), and indeed the authors provide an excellent summary of these limitations in the discussion (L406 onwards).

I appreciate the mathematical motivations for this term, and am happy for the authors to present the quadratic cost function based on purely mathematical motivations. However it is not sufficient to say the justification is ‘evident in many contexts’ - if the authors believe that the quadratic form has real-world justification in this context then they should expound upon this explicitly. Otherwise I think it would be best to make clear at this point in the paper that this functional form is purely mathematically motivated and may not have clinical relevance in this context (as discussed further in the discussion). If the authors wish to draw connections to other applications, where such a cost function would be appropriate, then they are welcome to but this is not necessary.

Furthermore, I appreciate the acknowledgement of some of these issues in L250 (‘However, if the true costs include any linear component such as the financial cost of drug supply, the quadratic cost function will be least accurate at lower levels of control and cancer. Hence the very low cost associated with an extended period of low-level treatment may not be realistic’), and do see this as a limitation of the continuous control approach, especially given that clinical dosing protocols are typically based on N-discrete level dosing. It would be nice to see these continuous protocols mapped onto an N-discrete level dosing protocol, to determine whether it is still effective. While I expect the authors may see the implementation of this as beyond the scope of the current paper, they could explore this suggestion within the discussion of future work (particularly within the context of clinically realistic treatment schedules - i.e. L435).

Finally, further comparison of the parameter values between the full model and the Null Model would be helpful. The small modifications of the Proliferation rate of P from 0.28 to 0.27 (and the death rate for P changed from 0.048 to 0.05), seem odd, and it is not clear why the simplification of the N population necessitated these changes. Were these changes introduced to maintain balanced competition with the N cells (which are less fit under the Null-N model), to avoid one cell population dominating? And were these values determined numerically, or through any analytic method to balance the fitness of the N and P populations?

Reviewer #2: The authors determined that the vast majority of my major comments were out of the scope of their current study. I certainly respect their judgement, but don't have anything new to add on the current study. Please refer to my original review for my thoughts on their paper. I appreciate that they fleshed out their figure captions to make the paper more digestible.

**Have the authors made all data and (if applicable) computational code underlying the findings in their manuscript fully available?**

Reviewer #1: Yes

Reviewer #2: Yes

PLOS authors have the option to publish the peer review history of their article (what does this mean?). If published, this will include your full peer review and any attached files.

Reviewer #1: **Yes: **Kit Gallagher

Reviewer #2: No

---

## [Editor Report · Acceptance letter]

PCOMPBIOL-D-24-00948R1

Optimal control of Multiple Myeloma assuming drug resistance

and off-target effects

Dear Dr Lefevre,

I am pleased to inform you that your manuscript has been formally accepted for publication in PLOS Computational Biology. Your manuscript is now with our production department and you will be notified of the publication date in due course.

With kind regards,

Zsofia Freund
